# Complex priors and flexible inference in recurrent circuits with dendritic nonlinearities

**Benjamin S. H. Lyo**
Center for Neural Science
New York University
blyo@nyu.edu

**Cristina Savin**
Center for Neural Science, Center for Data Science
New York University
csavin@nyu.edu

## Abstract

Despite many successful examples in which probabilistic inference can account for perception, we have little understanding of how the brain represents and uses structured priors that capture the complexity of natural input statistics. Here we construct a recurrent circuit model that can implicitly represent priors over latent variables, and combine them with sensory and contextual sources of information to encode task-specific posteriors. Inspired by the recent success of diffusion models as means of learning and using priors over images, our model uses dendritic nonlinearities optimized for denoising, and stochastic somatic integration with the degree of noise modulated by an oscillating global signal. Combining these elements into a recurrent network yields a stochastic dynamical system that samples from the prior at a rate prescribed by the period of the global oscillator. Additional inputs reflecting sensory or top-down contextual information alter these dynamics to generate samples from the corresponding posterior, with different input gating patterns selecting different inference tasks. We demonstrate that this architecture can sample from low dimensional nonlinear manifolds and multimodal posteriors. Overall, the model provides a new framework for circuit-level representation of probabilistic information, in a format that facilitates flexible inference.

## 1 Introduction

The organization of past experience in the form of internal models is crucial for making sense of noisy and incomplete sensory information. The framework of Bayesian inference successfully formalizes this knowledge in terms of probabilistic priors, and perception and action as probabilistic inference. However, despite many successes in explaining behavioral (Mamassian et al., 2002; Li et al., 2003; Gerardin et al., 2010; Girshick et al., 2011; Wei & Stocker, 2012; Ma, 2012; Hahn & Wei, 2022; Angeletos Chrysaitis & Seriès, 2023) and neural (Zemel & Dayan, 1998; Ma et al., 2006; Fiser et al., 2010; Savin & Denève, 2014; Savin et al., 2014; Orbán et al., 2016; Echeveste et al., 2020; Aitchison et al., 2021) observations through the lens of Bayesian inference, the circuit implementation of such computations remains poorly understood. In particular, we know relatively little about how priors are represented and the mechanics of how they affect perception.

Brain priors come in many flavors. For instance, priors over faces are structurally complex and invariant across tasks within subjects, despite substantial across subject variability (Houlsby et al., 2013). Such structural priors reflect natural world statistics; they allow for efficient learning and powerful generalization across a range of tasks (Gerardin et al., 2010; Houlsby et al., 2013). In contrast, contextual priors are relatively short-lived and adapt to reflect the needs of the task at hand (Seriès & Seitz, 2013; Darlington et al., 2018). Due to these differences, it is possible for different types of priors to have different neural representations (Rullán Buxó & Savin, 2021) and be accessed through separate neural mechanisms during inference.

Neural representation of priors are generally assumed to be implicit, with the exact form reflecting different schools of thought on neural correlates of uncertainty (Ma et al., 2006; Fiser et al., 2010). One set of models take the view that priors are represented in the stimulus encoding population,

with more neural resources dedicated to higher probability stimuli (Fischer & Peña, 2011; Girshick et al., 2011; Ganguli & Simoncelli, 2014; Bredenberg et al., 2020). The (log) prior could also be encoded in a separate subpopulation, e.g. in the form of a probabilistic population code and then linearly combined with the (log) sensory evidence (Ma et al., 2006). These theoretical schemes focus largely on the probabilistic representation of a single, one-dimensional stimulus feature, and cannot be readily generalized to the high-dimensional structured priors needed for representing natural stimuli (Houlsby et al., 2013; Durrant et al., 2023). Such complex distributions can be represented in stochastic recurrent circuits whose activity over time encodes samples from the posterior distributions (Hoyer & Hyvärinen, 2002; Fiser et al., 2010), or—in the limit of vanishing sensory evidence— samples from the prior (Berkes et al., 2011). In this neural sampling framework, prior information is implicitly encoded in the recurrent collaterals between the neurons that encode the inferred latents. However, the worked out examples of neural sampling dynamics have so far been restricted to priors and posteriors with relatively simple structure, e.g. Gaussian scale mixtures to account for statistics of gratings in early vision (Orbán et al., 2016), or multimodal distributions to model bistable percepts (Reichert et al., 2011; Savin et al., 2014). Finally, some priors could be encoded downstream of sensory representations (Ma, 2012), affecting perception through feedback mechanisms akin to attention (Whiteley & Sahani, 2012), although the circuit implementation of such effects remains underspecified. Overall, we are still missing circuit level models of probabilistic computation that capture the statistical complexity of natural priors and their flexible use across tasks.

The problem of representing complex high-dimensional distributions has close correspondents in machine learning. For example, digital images lie in a high dimensional pixel space, yet not all points in this space correspond to veridical images. Within this huge ambient dimensionality, naturalistic images trace low dimensional nonlinear manifolds, whose local coordinates represent continuous deformations or intensity variations. Among recent machine learning attempts at modeling such structure, diffusion models have emerged as particularly expressive and stable generative neural network models (Sohl-Dickstein et al., 2015; Kadkhodaie & Simoncelli, 2020; Ho et al., 2020), outperforming previous generative solutions such as VAEs, normalizing flows and GANs on the task of image synthesis (Dhariwal & Nichol, 2021). Their training involves local denoising operations, which can translate into relatively simple local learning objectives (Raphan & Simoncelli, 2011). Furthermore, they can be combined flexibly with different likelihoods to support inference (Sohl-Dickstein et al., 2015; Kadkhodaie & Simoncelli, 2021; Song & Ermon, 2019; Nichol & Dhariwal, 2021). Taken together, these features make diffusion models a valuable source of inspiration for building new classes of neural circuits for probabilistic computation.

Here we adapt several ideas developed in the context of diffusion models to construct a recurrent circuit model that can implicitly represent sensory priors and combine them with other sources of information to encode task-specific posteriors. Our solution relies on dendritic nonlinearities, optimized for denoising, and stochastic somatic activity modulated by a global oscillation that determines the effective rate of sampling. Additional inputs into the circuit which provide sensory or top-down contextual information shape the dynamics to generate samples from the corresponding posteriors. In simulations, we demonstrate several scenarios of prior and posterior encoding, including nonlinear manifolds embedded in a higher dimensional ambient space as priors and several likelihoods corresponding to bottom-up and top-down evidence. We also identify potential neural signatures of such probabilistic inference that might be testable experimentally. A software implementation of the model is available at `https://github.com/Savin-Lab-Code/LyoSavin2023`.

## 2 ENCODING COMPLEX PRIORS IN RECURRENT NEURAL CIRCUITS

**Representing priors with diffusion models.** Diffusion models (DMs) are generative neural networks that develop an implicit representation of data statistics by learning to denoise noisy versions of the data across multiple noise levels (Sohl-Dickstein et al., 2015; Kadkhodaie & Simoncelli, 2020; Ho et al., 2020). While there is more than one way to describe DMs, the essence of the process involves a predetermined *forward process*, which iteratively adds noise with known statistics to the input, and a learned *reverse process* which aims to "undo" the effects of the added noise.

Formally, the forward process transforms $N$-dimensional samples from a data distribution $\{\mathbf{x}_0\} \sim p(\mathbf{x}_0)$, e.g. natural images, by progressively adding small amounts of Gaussian noise over a number of steps $\Lambda$ following a pre-determined variance schedule $\{\sigma_\lambda\} \in (0, 1)$ that increases in magnitude

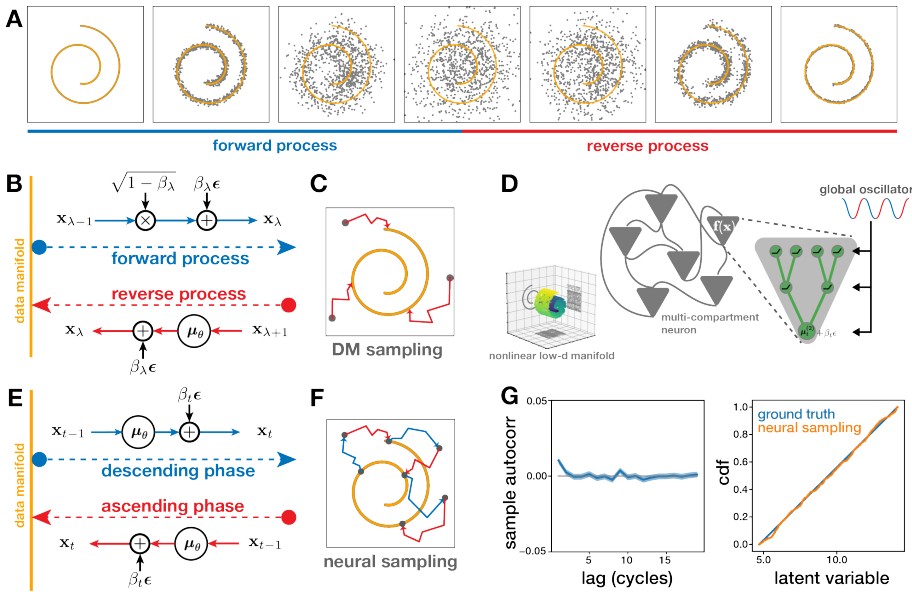

Figure 1: Diffusion models and their circuit implementation. **(A)** Simulated diffusion example for a two dimensional data distribution (yellow). Data samples from the manifold (gray dots) are corrupted by increasing levels of noise (blue), while the reverse process pushes the samples back onto the data manifold by recursively removing some of the noise (red). **(B)** Graphic summary of the two core DM operators: forward process adds noise with a predefined variance; dataset-specific reverse process removes the noise to bring the output back to the image manifold. **(C)** DM sampling starts from random points in the state space (gray dots), drawn independently from a standard Gaussian. The reverse process pushes these points back onto the manifold, resulting into independent samples from the data distribution. **(D)** Cartoon of a neural circuit encoding a nonlinear low dimensional manifold embedded in a larger neural feature space ("swiss-roll"). The network includes all-to-all recurrent connections between multi-compartment neurons (gray triangles), each consisting of a dendritic tree with optimized weights and ReLu nonlinearities, modulated by a global oscillatory signal. **(E)** Graphic summary of sampling in the neural circuit: starting from a valid point on the manifold, increasing amounts of noise due to the neural dynamics push the network off manifold, followed by the regular reverse process whose attractor forces brings the dynamics back on manifold, resulting in a new (potentially correlated) sample from the prior. **(F)** Neural sampling alternates between a descending phase which pushes points off the manifold through an increase in noise (blue trajectories), and an ascending phase, where attractor dynamics draw samples towards the manifold (red trajectories). **(G)** Numerical validation of neural sampling for the swiss-roll example. Left: autocorrelation function of the neural circuit generated samples. Right: marginal density recovery along the uniform dimension.

with index $\lambda = 1, \cdots, \Lambda$. Across these steps, noise gradually accumulates to ultimately overwhelm the signal, and the image distribution morphs into a standard $N$-dimensional multivariate Gaussian (Fig. 1A, B; blue). Each step of this process corresponds to the application of a stochastic forward transition operator that maps noisy versions of the data sample $\mathbf{x}_{\lambda-1}$ into noisier versions $\mathbf{x}_\lambda$,

$$p(\mathbf{x}_\lambda \mid \mathbf{x}_{\lambda-1}) = \mathcal{N}(\mathbf{x}_\lambda; \sqrt{1 - \sigma_\lambda}\mathbf{x}_{\lambda-1}, \sigma_\lambda I), \tag{1}$$

Analogously, the reverse process involves a series of applications of a Gaussian reverse transition operator that partially denoises noisy versions of the image $\mathbf{x}_{\lambda+1}$ into less noisy versions $\mathbf{x}_\lambda$ (Fig. 1A, B; red),

$$p_\theta(\mathbf{x}_\lambda \mid \mathbf{x}_{\lambda+1}) = \mathcal{N}(\mathbf{x}_\lambda; \boldsymbol{\mu}_\theta(\mathbf{x}_{\lambda+1}, \lambda), \sigma_\lambda I). \tag{2}$$

The mean of the reverse transition operator $\boldsymbol{\mu}_\theta(\mathbf{x}_{\lambda+1}, \lambda)$ is parameterized by a feedforward neural network that takes as input the noisy image from the previous iteration $\mathbf{x}_{\lambda+1}$ and the current noise index $\lambda$, with a matching, but reversed, variance schedule. The network learns this iterative mapping by optimizing a denoising objective that minimizes the reconstruction error, or equivalently, the L2

distance between the estimated and true noise present in the image:

$$\mathcal{L} = \mathbb{E}_{\lambda \sim [1,\Lambda], \mathbf{x}_0, \boldsymbol{\epsilon}_\lambda} \left[ \|\tilde{\boldsymbol{\mu}}_\lambda(\mathbf{x}_\lambda, \boldsymbol{\epsilon}_\lambda) - \boldsymbol{\mu}_\theta(\mathbf{x}_\lambda, \lambda)\|_2^2 \right] \tag{3}$$

$$= \mathbb{E}_{\lambda \sim [1,\Lambda], \mathbf{x}_0, \boldsymbol{\epsilon}_\lambda} \left[ \|\boldsymbol{\epsilon}_\lambda - \boldsymbol{\epsilon}_\theta(\mathbf{x}_\lambda, \lambda)\|_2^2 \right], \tag{4}$$

where $\boldsymbol{\epsilon}_\lambda \sim \mathcal{N}(0, I)$, $\tilde{\boldsymbol{\mu}}_\lambda$ is the mean of the forward transition operator at step $\lambda$ when conditioned on a clean image $\mathbf{x}_0$, and $\boldsymbol{\epsilon}_\theta$ is the network's estimate of the noise present in the noisy image. This objective is equivalent to learning the flow field associated with the score of the reverse transition operator at every noise level $\nabla_{\mathbf{x}_\lambda} \log p(\mathbf{x}_\lambda \mid \mathbf{x}_{\lambda+1})$ (Hyvärinen, 2005; Vincent, 2011). At the end of training, the model can generate novel sample images from the same distribution, by starting with i.i.d. multivariate Gaussian noise and iteratively applying the reverse transition operator following the same variance schedule. Each run of this procedure generates a new independent image sample.

**A diffusion-based circuit for sampling-based probabilistic computation.** Several properties of diffusion models seem appealing from a biological perspective. In particular, the iterative nature of the operations make them naturally map into recurrent circuit dynamics. In this view, the data distribution follows a nonlinear manifold attractor, embedded in the $N$-dimensional space of sensory features encoded by the network (not necessarily image pixels). The forward process injects noise that pushes data samples away from the manifold, generating signals for learning, while the reverse process implements denoising. Here, we chose to focus specifically on the circuit dynamics and the mechanism for sample generation. We leave the learning aspect of the problem for future work.

A recurrent circuit can implement something akin to the DM reverse process, by assuming that the the nonlinear reverse operator $p_\theta(\mathbf{x}_\lambda \mid \mathbf{x}_{\lambda+1})$ is computed in the dendritic trees of a set of $N$ cortical pyramidal neurons. At each time step, nonlinear dendritic computations (Poirazi et al., 2003; Polsky et al., 2004; Jadi et al., 2014; Ujfalussy et al., 2018) yield somatic currents that reflect the mean of the Gaussian transition operator $\boldsymbol{\mu}_\theta$, which is subsequently corrupted by somatic noise. As an analogue for the variance schedule, we introduce a global time-varying signal $\beta_t \in (0, 1)$ which modulates the nonlinear operation of the dendrites and the magnitude of somatic noise. Mechanistically, this type of modulation could be implemented by local inhibitory subunits (Murayama et al., 2009; Roux & Buzsáki, 2015). We adopt the convention of associating increasing $\beta_t$ with a reduction in noise, although this choice is somewhat arbitrary. Scalar $\beta_t$ is a sinusoid which fluctuates over time between 0 and 1, in anti-phase to the DM noise schedule $\sigma_\lambda$, to account for the fact that downstream circuits read out information at the peak of the local oscillation (Savin et al., 2014):

$$\beta_t = \frac{1}{2}\left(1 + \cos\left(\frac{2\pi t}{T}\right)\right) = 1 - \frac{\sigma_\lambda - \sigma_1}{\sigma_\Lambda - \sigma_1}, \tag{5}$$

where $T$ is the number of time steps in a cycle. Putting everything together, recurrent circuit dynamics take the form : $\mathbf{x}_{t+1} = f(\mathbf{x}_t, \beta_t) + (1 - \beta_t)\boldsymbol{\epsilon}_t$, where $f(\cdot, \cdot)$ reflects the nonlinear function computed by the dendrites, itself modulated by the global oscillation, and $\boldsymbol{\epsilon}_t \sim \mathcal{N}(0, I)$ are i.i.d. samples from an isotropic Gaussian. Following Poirazi et al. (2003), dendritic computations are modeled using a tree-structured artificial neural network (ANN), whose parameters we train via gradient descent using the standard DM denoising objective (Fig. 1D). Unlike previous ANN-based models of dendritic processing, we use ReLU nonlinearities to facilitate training, with the shape of the dendritic tree parameterized by its depth and branching factors at each level.[1] This biologically-motivated solution represents an architectural departure from standard DMs, which assume all-to-all connectivity. The ability to represent priors is relatively robust to variations in tree geometry, although learning speed can vary with architecture. In particular, varying tree depth and width while keeping the total number of parameters constant identifies shallow architectures as the fastest learners in our example (see Suppl. B.1). This implies that one can determine an optimal architecture for encoding any given prior, with potential implications for cortical dendritic morphology.

Another key distinction between traditional DMs and biology is the initial condition for the reverse process during sampling. In DMs, initial conditions are drawn independently from a standard Gaussian, which would yield unrealistic state jumps in the circuit dynamics. In contrast, in the recurrent circuit the same neural units need to support the dynamics at all times. As a result, we use the same *neural transition operator*, $p_\theta(\mathbf{x}_t \mid \mathbf{x}_{t-1}) := \mathcal{N}(\mathbf{x}_t; \boldsymbol{\mu}_\theta, (1 - \beta_t)I)$, but under an increasing

---

[1]Unless otherwise specified, simulations use a depth of 7 and branching factor of 3, except in the most proximal section, which has a branching factor 4.

noise schedule to reinitialize the reverse process and generate another sample from the prior (Fig. 1E). If we were to use the DM forward process for this operation, the end effect would be statistically equivalent to the original (since noise dominates the signal, by construction, the initial conditions are forgotten). The proposed neural dynamics do not necessarily come with the same mathematical guarantees, but they do significantly simplify the circuit's operations. In this scheme, sampling from the prior entails running the same stochastic nonlinear recurrent dynamics, at a pace set by an oscillatory $\beta_t$ signal, with one new sample from the prior generated once per cycle (Fig. 1F). [2] The precise structure of these oscillations can be relaxed to account for frequency and amplitude fluctuations seen biologically (Suppl. Sec. B.5).

We numerically tested the quality of the samples generated by our neural circuit in a toy example of a two-dimensional nonlinear manifold (shaped as a "swiss-roll", see Fig. 1D inset) with linear dimensionality 3, embedded in an ambient feature space with dimensionality $N = 10$. This simple example both captures the essence of priors of interest (i.e., nonlinear low-dimensional manifold embedded in a larger feature space) and is simple enough to exploit its structure when assessing the quality of the generated samples, something which is generally difficult for complex distributions (Theis et al., 2016). First, we tested the effects of the neural approximation on the autocorrelation function of the generated samples and found that this remains steadily around zero even with the shortest lags (Fig. 1G, left), proving that, despite the approximation, the generated samples are essentially independent. Second, to test the variability of the generated samples we numerically estimated the density along the curved axis of the manifold, which—by construction—has a uniform distribution. We found that the empirical density constructed using the neural samples closely matched the ground truth (Fig. 1G). This is also true for the marginal along the orthogonal dimension (Suppl. B.2). While the quality of samples is harder to estimate, we also find good quality representations of a high dimensional prior trained on the MNIST dataset (Deng, 2012) (see Suppl. B.6). Overall, these results indicate that the constraints imposed by biology may have a minimal effect on the quality of neural sampling as compared to traditional DMs.

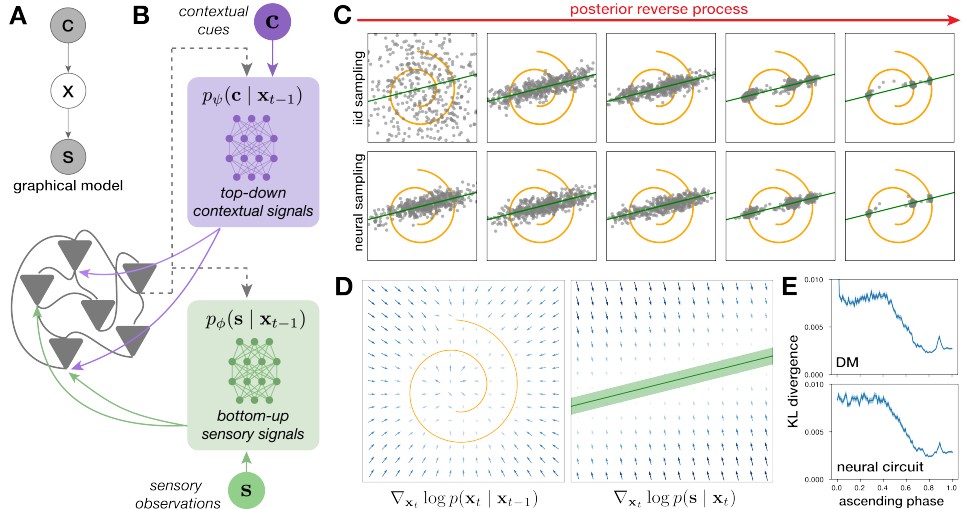

Figure 2: Posterior inference. **(A)** Graphical model illustrating dependencies between latent variables $\mathbf{x}$ and sensory inputs $\mathbf{s}$. **(B)** Extended neural circuit for flexible inference: prior dynamics are reshaped by external signals due to either bottom-up sensory observations (green) and/or top-down contextual cues (purple). These inputs are themselves modulated by the state of the circuit, through inter-area interactions. **(C)** Snapshots of posterior samples during the reverse process for the diffusion model (top) and the neural circuit (bottom) under the influence of bottom-up sensory signals. **(D)** Flow fields corresponding to a prior (left) and a linear gaussian sensory likelihood (right; shaded green area 1sd). **(E)** Quantification of the cost of approximating the likelihood when sampling using DM (top) or neural circuit (bottom) dynamics, measured by the KL divergence between the true posterior (evaluated at $\boldsymbol{\mu}_\theta$) and the approximate posterior (evaluated at $\mathbf{x}_{t-1}$).

---

[2] In all simulations we initialize the dynamics randomly and discard the first cycle as burn-in period.

## 3 FLEXIBLE INFERENCE

**Using the same prior-encoding circuit for inference.** Machine learning has proposed several algorithmic solutions for combining DM-based prior representations with different likelihood functions, with practical applications such as in-painting, super-resolution, compressed sensing, or class-specific conditioning (Sohl-Dickstein et al., 2015; Dhariwal & Nichol, 2021; Kadkhodaie & Simoncelli, 2021; Song & Ermon, 2019; Ho & Salimans, 2021). In order to maximize the flexibility of our circuit implementation, we take advantage of a modular approach that reuses the dynamics that sample from the prior to do flexible inference. In particular, posterior samples are constructed by altering the flow-fields of the DM-prior, in the form of state-dependent additive changes to the DM reverse transition operator (Sohl-Dickstein et al., 2015; Dhariwal & Nichol, 2021).

In general, our approach entails defining a *posterior neural transition operator* which reflects how additional evidence changes the beliefs about the represented latent variables (Fig.2A). Formally, posterior sampling dynamics take the form (see Suppl. A.1 for derivation):

$$p(\mathbf{x}_t \mid \mathbf{x}_{t-1}, \mathbf{y}) := \mathcal{N}(\boldsymbol{\mu}_\theta(\mathbf{x}_{t-1}) + \gamma(1 - \beta_t)\mathbf{g}_t(\mathbf{x}_{t-1}, \mathbf{y}), (1 - \beta_t)I), \tag{6}$$

where the effects of the prior are reflected in the mean $\boldsymbol{\mu}_\theta$, as before, but shifted by the (log)likelihood gradient $\mathbf{g}_t(\mathbf{x}_{t-1}, \mathbf{y}) = \nabla_{\mathbf{x}_{t-1}} \log p_\phi(\mathbf{y} \mid \mathbf{x}_{t-1})$, which, unlike Dhariwal & Nichol (2021), is evaluated with respect to a noisy data sample at the previous time step $\mathbf{x}_{t-1}$. Scalar $\gamma$ is a hyperparameter that weighs the relative contribution of the log prior and log likelihood. This hyperparameter is set to 1 in Sohl-Dickstein et al. (2015) and varied in Dhariwal & Nichol (2021) to optimize a trade-off between sample fidelity and diversity (here, $\gamma = 1$). The relative influence of the additional evidence on the circuit dynamics varies across the cycle, scaled by $(1 - \beta_t)$. Mechanistically, this solution means that the local recurrent circuit representing the prior receives an external input drive, itself gated by the global oscillation, which shifts circuits dynamics to sampling from the corresponding posterior. The nature of the input signal is specific to each of the sources of evidence, and implemented in a separate feedforward sub-circuit, which interacts bi-directionally with the primary recurrent network. At each timestep, the sub-circuit receives the activity of the recurrent network from the previous timestep and returns an input drive that is added to the somatic current of each neuron in the recurrent network. Functionally, this sub-circuit implements the computation of $\mathbf{g}_t(\mathbf{x}_{t-1}, \mathbf{y})$ (Fig. 2B). The functional form of this circuit is problem specific, determined either in closed form (see sensory inference example below), or itself learned (e.g., in the context cue example). Different sources of evidence can operate in parallel by providing their own additive contribution to the circuit drive.

**Inferring latents from sensory evidence.** As a first concrete example of inference, we show a simplified version of a sensory perception task in which the input is only partially observed and the corresponding features need to be inferred given this incomplete observation (as would be the case in occlusion, for instance). In particular, we take the sensory observation to induce a linear constraint on the latent variable ($N = 2$ for easy visualization), $\mathbf{M}_s^\top(\mathbf{x} - \mathbf{x}_c) = 0$, parameterized by a unit norm linear operator $\mathbf{M}_s$ and offset $\mathbf{x}_c$, with Gaussian-distributed uncertainty around it (Fig.2B, right). Formally, the corresponding likelihood score is orthogonal to the constraint manifold and linearly rescaled by distance to it, $\mathbf{g}_t(\mathbf{x}_{t-1}, s) = \frac{1}{\sigma_s^2}\mathbf{M}_s\mathbf{M}_s^\top(\mathbf{x}_{t-1} - \mathbf{x}_c)$; parameter $\sigma_s$ defines the degree of sensory uncertainty. As was the case for the prior, the DM implementation of this inference problem generates samples starting from the isotropic $N$-dimensional Gaussian, with the posterior reverse dynamics bringing the samples towards the posterior, here concentrated at the intersection of the two manifolds corresponding to the prior and likelihood, respectively (Fig. 2C, top). The neural posterior dynamics find a similar solution (Fig. 2C, bottom), but with initial conditions reflecting the likelihood. This is due to the sequential sampling process which has the same dynamics operating in both forward and reverse phases of the oscillation, and to the fact that the likelihood is gated oppositely to the prior (due to the $1 - \beta_t$ scaling, see Eq. 6).

When it comes to inference, the sampling dynamics of the neural circuit approximate DM posterior dynamics in two important ways. First, as for the prior, they introduce sequential effects due to the continuous use of the same operator across all phases of the cycle. Second, the computation of the likelihood is itself approximate, as $\mathbf{g}_t$ is evaluated at point $\mathbf{x}_{t-1}$ instead of the $\boldsymbol{\mu}_\theta(\mathbf{x}_{t-1})$ required in the original posterior flow field derivation (Sohl-Dickstein et al., 2015; Dhariwal & Nichol, 2021). This switch was needed to account for temporal causality and the fact that the sensory area providing this input does not have access to local somatic currents $\boldsymbol{\mu}_\theta(\mathbf{x}_{t-1})$. In principle, both effects should

diminish with a finer discretization of time, but we also numerically quantified the effects of each approximation individually (Fig. 2E, top) and together (Fig. 2E, bottom). In particular, we collected samples generated according to either the DM reverse operator (using $\boldsymbol{\mu}_\theta(\mathbf{x}_{t-1})$) or the neural posterior sampler (using $\mathbf{x}_{t-1}$). We quantified the similarity of the resulting sample distributions as a function of $\beta_t$, using a naive KL divergence estimator (10000 samples each, with space discretized into 225 bins). We found that discrepancies between the distributions are largest at the start of the reverse process. In this regime, the noise and flow fields are large leading to substantial differences between $\mathbf{x}_{t-1}$ and $\boldsymbol{\mu}_\theta(\mathbf{x}_{t-1})$ (Fig. 2E, top). In principle, this might lead to catastrophic accumulation of errors and large sampling biases; however, the attractor dynamics prevent this from happening. The effects of the approximation reduce over the cycle, with the KL divergence approaching zero. We see largely the same effects when the neural likelihood approximation is used in conjunction with the neural prior approximation (same operator throughout; Fig. 2E, bottom). Overall, these numerical results suggest that, at least in the simple toy examples considered here, the effects of neural approximations on inference are minimal.

**Inference with multiple sources of information.** The main benefit of separating the neural substrates implementing the prior from the computation of the likelihood is that it makes inference flexible. Gating signals from different sources of evidence allows the model to reuse the same prior information across multiple tasks. Beyond the sensory example above, another important source of evidence that shapes perception/inference comes from contextual priors, reflecting expectations about which of the latent features are likely to occur (e.g. attention). To illustrate this idea, we introduce a discrete context random variable, $c$, whose value leads to different expectations about the latent variable $\mathbf{x}$, formally making the prior a mixture of 1D manifolds, with data uniformly distributed along each (Fig. 3A, top).[3] This additional complexity does not affect the neural dynamics sampling from this prior (Fig. 3A, bottom), but allows us to demonstrate richer inference scenarios.

Sampling from the context conditioned posterior $\mathrm{p}(\mathbf{x}|c)$ proceeds very similarly to the sensory inference example above, but the drive to the circuit is given by a separate top-down feedforward sub-circuit which implements a different $\mathbf{g}_t(\mathbf{x}_{t-1}, c) = \nabla_\mathbf{x} \log p_\phi(c \mid \mathbf{x}_{t-1})|_{\mathbf{x}=\mathbf{x}_{t-1}}$. [4] The corresponding flow field and posterior samples are shown in (Fig. 3B), where we see the dynamics being strongly attracted to the region of the latent space corresponding to the cued context (the 'S' manifold). Finally, top-down contextual signals can be combined with the bottom-up sensory evidence (Fig. 3C) to jointly constrain the posterior distribution from which we draw samples (Fig. 3D). When multiple sources of evidence, $y^{(i)}$, are available, their net effect on the likelihood combines additively, $\mathbf{g}_t(\mathbf{x}_{t-1}, \mathbf{y}) = \sum_i \gamma_i g_t^{(i)}(\mathbf{x}_{t-1}, y^{(i)})$. This modular parametrization is at the root of the circuit's inferential flexibility, with individual components gated in or out depending on availability of information and current task demands (Womelsdorf et al., 2014; Kuchibhotla et al., 2017).

## 4 NEURAL SIGNATURES

One unique feature of the inferential process proposed here is how the phase of the circuit oscillation modulates the relative contribution of different inputs to the circuit. At one extreme ($\beta_t = 0$), the effect of the local attractor dynamics is weak and the corresponding prior unstructured, with neural activity dominated by the likelihood component. At the other extreme ($\beta_t = 1$), the effect of the likelihood on the dynamics is weak, and neural activity corresponds to samples from the posterior (Fig. 4A; see also Eq. 6). This dialing up and down of different sources of evidence opens the door for experimental validation of the model. Concretely testing such dynamics involves assessing patterns of population activity during perception, at different phases of the dominant circuit oscillation, as samples from the intermediate posterior. Samples from the prior $\mathrm{p}(\mathbf{x})$ can be obtained from spontaneous activity, in the absence of a sensory stimulus and outside a specific task context, while a reasonable measure of the effects of the likelihood is given by the activity at the lowest phase of the posterior dynamics, $\mathrm{p}(\mathbf{x}_t|\mathbf{y}, \beta_t = 0)$. Given these statistics, the change in the contribution of the likelihood and the prior on this posterior can be estimated as the KL divergence between the activity of the circuit during inference, and these two reference distributions. The precise assessment of differences between potentially complex and structured distributions based on

---

[3]The ambient space is 2D, for simplicity.

[4]Note here that the direction of the conditioning is switched from the natural causal one, which means that this map can be learned from labeled examples by optimization (see Suppl. A.1 for derivation).

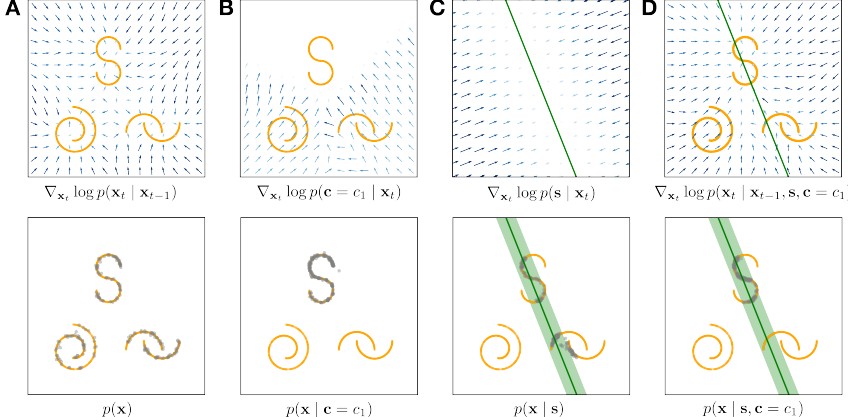

Figure 3: **(A)** Flow fields (top) and samples (bottom) from an example prior mixture distributions, with individual components uniform distributions along low dimensional nonlinear manifolds.**(B)** Likelihood flow field and samples from the corresponding context-conditioned posterior. **(C)** Likelihood flow field and samples from the corresponding sensory conditioned posterior. **(D)** Posterior flow field and samples from posterior conditioned on both sensory evidence and context.

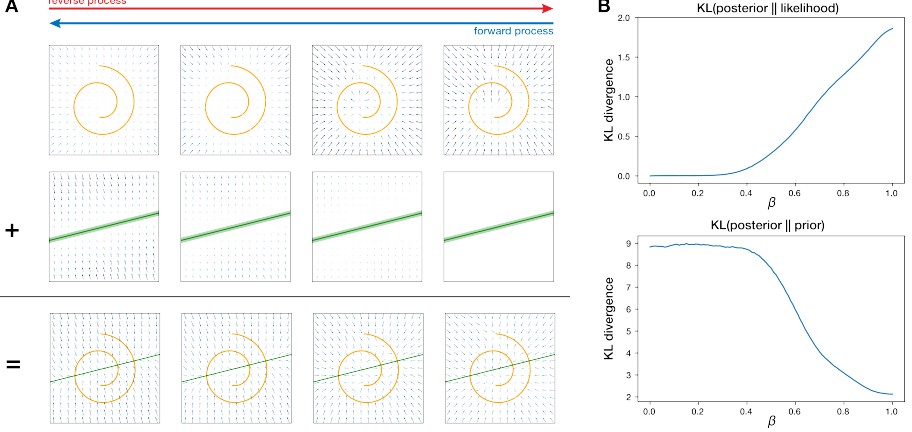

Figure 4: **(A)** Relative contribution of the prior and likelihood scores during the neural forward and reverse processes. **(B)** Fluctuating similarity of neural activity to the likelihood (top) and the prior (bottom) over the ascending phase of the oscillation $\beta_t$. Distance between sampled distributions measured by a naive histogram-based KL divergence estimator.

empirical measurements may seem impractical, but there is precedent in the literature for estimating KL divergences in population activity statistics using coarse discretizations of the space (e.g. temporal binning and thresholding of multiunit activity) (Berkes et al., 2011). For simplicity, here we draw $10^5$ from each distribution, discretize the latent space into $400$ bins and compute a naive estimate just based on the resulting histograms, although more sample efficient estimators are certainly available (Nemenman et al., 2001). Despite the coarseness of the measure, the intuition about the effects of $\beta_t$ holds: population activity statistics become increasingly similar to the prior, and dissimilar from the likelihood, over the ascending phase of the oscillation (Fig. 4B). With a more precise understanding of the mechanics of gating different type of signals into the circuit, one could further envision causal manipulations that precisely bias the outcome in one or the other direction.

## 5 DISCUSSION

Natural sensory inputs are often structured into complex nonlinear manifolds in high dimensions (Carlsson et al., 2008; Fefferman et al., 2016; Gorban & Tyukin, 2018; Brown et al., 2023). How

does the brain represent such prior knowledge and uses it across tasks? Conceivably, one could solve this problem by learning a mapping between stimuli and responses *de novo* for each task. However, a more efficient representation would consolidate and leverage universally useful information across tasks. Here, we have proposed a circuit implementation of across-task inference in which a common prior is encoded in the form of a recurrent neural circuit with dendritic nonlinearities optimized for denoising. The stochastic dynamics of this circuit provide samples from the prior at the peak of an ongoing local oscillation, while additional inputs (carrying sensory or contextual information) switch the dynamics to sampling from the corresponding posterior. This solution offers the first circuit model for reusing priors across tasks, as seen behaviorally (Houlsby et al., 2013), and has measurable neural signatures based on neural population recordings in animals.

Theories of probabilistic brain computation based on neural sampling suffer from two main practical challenges. The first is time, as estimating posterior expectations for decision making requires integrating samples over time (Rullán Buxó & Savin, 2021). This makes the speed of sampling a key constraint on plausible neural sampling dynamics (Hennequin et al., 2014; Savin & Denève, 2014; Aitchison & Lengyel, 2016; Masset et al., 2022). Second, although in principle sampling makes most sense for distributions with complex structure that cannot be well-represented parametrically (Fiser et al., 2010), in practice circuit models of sampling tend to be restricted to relatively simple distributions (Chen et al., 2023). The dendritic nonlinearities and DM-inspired oscillatory sampling schedule in our solution naturally overcome both limitations: by construction, the circuit is designed to sample from complex distributions, e.g. involving low dimensional nonlinear manifolds embedded in high dimensional spaces, and mixture distributions with low probability gaps between the components, which pose fundamental challenges for classic Markov Chain Monte Carlo approaches (Aitchison & Lengyel, 2016). Also by construction, DMs are designed to provide independent samples from the target distribution at the end of each reverse run, something that our neural circuit approximation seems to preserve to a large degree.

The specific role of oscillations in sampling resembles a previous model of sampling in the hippocampus (Savin et al., 2014) inspired by tempered transitions (Neal, 1996). In this view, $\beta$ functions as an inverse temperature annealing the target distribution so as to reduce the gap (and thus increase the likelihood of transitions) between modes; this may inform the theoretical analysis of the sampler's properties. Additionally, theta oscillations naturally set the tempo in the hippocampus, which may instruct the experimental validation of our proposal. The concrete approach for testing our experimental prediction was strongly influenced by Berkes et al. (2011), who originally used KL divergences to measure the calibration of (sampling based) probabilistic models of early vision over development. While segregating the responses by oscillatory phase brings additional data and estimation challenges, the Berkes result, together with the evidence of hippocampal theta phase sampling from Savin et al. (2014), provides reassurance that experimental validation of the model might be possible in practice.

While our proposed solution relies on nonlinearities in the dendrites as an implementation of the reverse operator, it is not clear that this is the only way to map DM operations to biology. In principle, the computations at each timestep could be carried out by a small feedforward subnetwork, recursively interacting between them. This version has the advantage of fewer restrictions on the architecture, but requires additional biological accommodations. In particular, for consistency noise would need to be distributed equally in all the neurons of the network (as opposed to just the last layer in DMs and our solution). Computationally, this solution remains viable (Suppl. B.4), although the architecture is arguably more contrived. Models of learning might better distinguish between these architectures in the future.

Here, we have purposefully abstracted away the nature of the features, so that the modeled circuit could be any processing stage in the cortical hierarchy, receiving evidence from one or several sensory domains bottom-up, and contextual top-down information from higher areas. This modular architecture opens the way for hierarchical probabilistic representations, by stacking such recurrent circuits with appropriate conditioning between them (Qiang et al., 2023). Such an architecture would also help limit the computational complexity of the neural nonlinearities involved in any stage: pixel-level DMs models of images are huge, with parameters on the order of billions (Ramesh et al., 2021; Saharia et al., 2022), but models of abstract image features are likely more tractable. Future work on hierarchical sampling neural circuits might lead not only to better accounts of brain representations but also to more compact machine learning models.

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

# A  Supplementary derivations

## A.1  Posterior sampling dynamics

We modify the original posterior flow field derivation by Sohl-Dickstein et al. (2015) to allow for a biologically plausible posterior sampling algorithm. We wish to derive the *posterior neural transition operator* $p(\mathbf{x}_t \mid \mathbf{x}_{t-1}, \mathbf{y})$ by combining the prior neural transition operator $p(\mathbf{x}_t \mid \mathbf{x}_{t-1})$ with an arbitrary likelihood signal $p_\phi(\mathbf{y} \mid \mathbf{x}_t)$. Recall that our prior neural transition operator is Gaussian:

$$p(\mathbf{x}_t \mid \mathbf{x}_{t-1}) = \mathcal{N}(\boldsymbol{\mu}_\theta, (1 - \beta_t)I)$$

$$\log p(\mathbf{x}_t \mid \mathbf{x}_{t-1}) = -\frac{1}{2(1 - \beta_t)}\|\mathbf{x}_t - \boldsymbol{\mu}_{\theta,t}\|^2 + C$$

where C is a constant. Under some mild assumptions, we can approximate the log likelihood $\log p_\phi(\mathbf{y} \mid \mathbf{x}_t)$ by Taylor expanding around $\mathbf{x}_t = \boldsymbol{\mu}_{\theta,t}$, where $\boldsymbol{\mu}_{\theta,t} = \boldsymbol{\mu}_\theta(\mathbf{x}_{t-1})$:

$$\log p_\phi(\mathbf{y} \mid \mathbf{x}_t) \approx \log p_\phi(\mathbf{y} \mid \mathbf{x}_t)\big|_{\mathbf{x}_t=\boldsymbol{\mu}_{\theta,t}} + (\mathbf{x}_t - \boldsymbol{\mu}_{\theta,t})\nabla_{\mathbf{x}_t} \log p_\phi(\mathbf{y} \mid \mathbf{x}_t)\big|_{\mathbf{x}_t=\boldsymbol{\mu}_{\theta,t}}$$

$$= (\mathbf{x}_t - \boldsymbol{\mu}_{\theta,t})\mathbf{g}_t(\mathbf{x}_{t-1}, \mathbf{y}) + C_1$$

where the likelihood flow field $\mathbf{g}_t(\boldsymbol{\mu}_{\theta,t}, \mathbf{y}) = \nabla_{\mathbf{x}_t} \log p_\phi(\mathbf{y} \mid \mathbf{x}_t)\big|_{\mathbf{x}_t=\boldsymbol{\mu}_{\theta,t}}$ and $C_1$ is a constant. Note that in this derivation, the gradient of the log likelihood is evaluated at $\boldsymbol{\mu}_\theta(\mathbf{x}_{t-1})$, the local somatic current at the current time step. This makes a direct biological interpretation of the model problematic since the sensory area providing this likelihood signal cannot access the somatic current and only has access to information from the previous time step. In order for our model to obey temporal causality and local access to information, we suppose instead that the likelihood flow field receives as input a point $\mathbf{x}_{t-1}$ from the previous time step which is used to evaluate the likelihood flow field $\mathbf{g}_t(\mathbf{x}_{t-1}, \mathbf{y}) = \nabla_{\mathbf{x}} \log p_\phi(\mathbf{y} \mid \mathbf{x}_{t-1})|_{\mathbf{x}=\mathbf{x}_{t-1}}$. This approximation becomes more exact the closer we are to the data manifold: since $\boldsymbol{\mu}_\theta(\mathbf{x}_{t-1})$ can be thought of as the prior network's current guess of a point on the prior manifold, the error from substituting $\mathbf{x}_{t-1}$ for $\boldsymbol{\mu}_{\theta,t}$ goes to zero as $\mathbf{x}_t$ approaches the prior manifold. We demonstrate this numerically in Fig. 2E.

The posterior neural transition operator combines the prior transition operator and likelihood signal using Bayes rule:

$$\log p(\mathbf{x}_t \mid \mathbf{x}_{t-1}, \mathbf{y}) = \log p_\theta(\mathbf{x}_t \mid \mathbf{x}_{t-1}) + \log p_\phi(\mathbf{y} \mid \mathbf{x}_t) + C_1$$

$$= -\frac{1}{2(1 - \beta_t)}\|\mathbf{x}_t - \boldsymbol{\mu}_{\theta,t}\|^2 + (\mathbf{x}_t - \boldsymbol{\mu}_{\theta,t})\mathbf{g}_t + C_2$$

$$= -\frac{1}{2(1 - \beta_t)}\|\mathbf{x}_t - \boldsymbol{\mu}_{\theta,t} - (1 - \beta_t)\mathbf{g}_t\|^2 + \frac{\mathbf{g}_t^2(1 - \beta_t)}{2} + C_2$$

$$= -\frac{1}{2(1 - \beta_t)}\|\mathbf{x}_t - (\boldsymbol{\mu}_{\theta,t} + (1 - \beta_t)\mathbf{g}_t)\|^2 + C_3$$

$$= \log \mathcal{N}(\mathbf{x}_t; \boldsymbol{\mu}_{\theta,t} + (1 - \beta_t)\mathbf{g}_t, (1 - \beta_t)I) + C_4$$

where $C_{1-4}$ are constants that do not contain $\mathbf{x}_t$. As we can see, the posterior transition operator is again a Gaussian and $C_4$ corresponds to its normalizing constant. The effect of the external likelihood signal is to shift the mean of the prior transition operator by an amount $(1 - \beta_t)\mathbf{g}_t$.

To sample from the context conditioned posterior $p(\mathbf{x} \mid c)$, we learn a map from $\mathbf{x}_t$ to $c$ by training a feedforward neural network to classify points $\mathbf{x}_t$ via optimization. For the trimodal prior in Fig. 3, this is done by assigning one of three class labels to samples from the corresponding mode. We construct the classifier training set by adding i.i.d. multivariate Gaussian noise of varying degrees of variance to clean samples from the prior using the DM forward process. As with the unimodal swiss-roll prior, at large degrees of noise most of the signal is removed and the distribution resembles a white Gaussian. The classifier network is then trained via backpropagation to receive $\mathbf{x}_t$ as input and identify the class of the corresponding clean data sample. Since the last layer of the classifier is a Softmax layer that normalizes the outputs to one, we interpret the values of the classifier output as the probability distribution $p(c \mid \mathbf{x}_t)$. $\mathbf{g}_t(\mathbf{x}_{t-1}, c)$ is evaluated using the torch autograd package.

# B SUPPLEMENTARY RESULTS

## B.1 EFFECTS OF DENDRITIC TREE MORPHOLOGY

We studied how the morphology of the dendritic networks affected the speed of learning by conducting a grid search over dendritic architectures that differed in width and depth. To isolate the effect of the architecture choice, we kept the total number of parameters roughly the same across all networks, meaning that as we increased network depth we also decreased network width.

A dendritic architecture can be fully specified by an array of branching factors, which denotes the number of children nodes at each layer. The array gives branch factors in order of distal to proximal layers. The six architectures are defined as follows:

| Depth | Morphology | Num. of parameters |
|---|---|---|
| 2 | [59, 59] | 35048 |
| 4 | [8, 8, 7, 7] | 33154 |
| 5 | [5, 5, 5, 5, 5] | 34372 |
| 6 | [4, 4, 4, 4, 4, 3] | 34814 |
| 7 | [3, 3, 3, 3, 3, 3, 4] | 34986 |
| 10 | [2, 2, 2, 2, 2, 2, 2, 2, 3, 3] | 32234 |

Table S1: Morphology of dendritic networks.

We trained four iterations of each network architecture with different random seeds. The networks were trained on the 2D swiss roll manifold using the Adam optimizer with a learning rate of 3e-4 over 1.5e6 epochs. We quantified the similarity of the network to the ground truth manifold as a function of training epoch using a naive KL divergence estimator (10000 samples each, with space discretized into 255 bins).

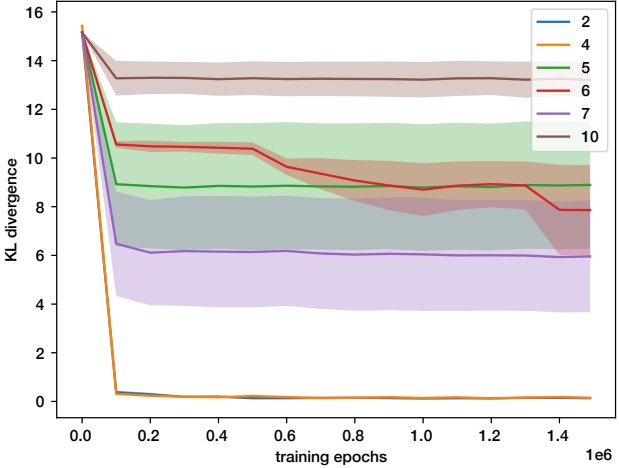

Figure S1: Network performance over training. Each line corresponds to the performance of a model architecture of a particular depth. We see that the KL divergence of shallower networks (depths 2, 4) decreased the fastest, while deeper network converges slower. Note that the speed of learning does not scale monotonically with network depth; for example, the 7-layer network learns faster than the 5 layer network.

We see that shallower but wider networks learned faster than deeper but thinner networks, and converged to a KL divergence value much lower than the other networks. However, we cannot conclude that shallower networks perform better overall: the KL divergence value for the 6 layer network even towards the end of the 1.5e6 training epochs indicates that the networks have not reached the global minimum of their loss functions, and suggests that with more training epochs the network performance may continue to improve.

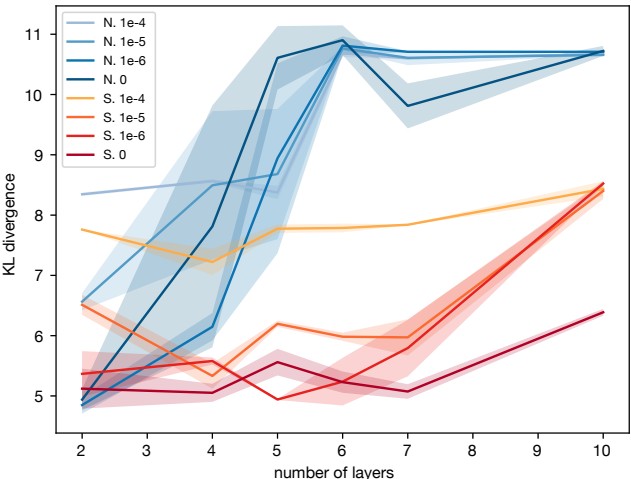

Figure S2: Effects of sparse recurrent connections targeting different locations on the dendritic tree. Quality of prior samples as a function of the dendritic tree depth. $S$ and $N$ in the legend indicate architectures with and without skip connections, respectively. Number denote the strength of the L1 regularization. We trained two models for each architecture. Shaded regions indicate variability in the KL divergence resulting from variability in generated samples.

One limitation of the architecture is that the resulting recurrent network has dense connectivity, linking each neuron to all distal branches of all neurons, which is biologically unrealistic. We adjusted the tree architecture so that inputs from other neurons can project to different locations on the target neuron's dendritic tree. Architecturally, this was achieved in the form of skip connections from the neurons in the first layer (representing the output of the somatic compartment at the previous timestep) to intermediate subunits in the dendritic arbor of all neurons. These skip connections, unlike those commonly found in deep neural network models (He et al., 2015), have learnable weights. The synaptic inputs from these skip connections are summed by the subunit together with incoming activity from more distal subunits. To encourage sparsity in the recurrent inter-neuron connectivity, we added L1 regularization on the synaptic strengths.

We studied prior representational quality of the biologically-refined version of the model while varying the depth of the dendritic tree for the same task (10 dimensional network representing a 3 dimensional "swiss-roll" latent manifold). All models were trained over 2e6 epochs with different degrees of L1 regularization. We compared model generated samples against the ground truth by calculating the KL divergence between the two distributions formed by projections onto a two dimensional cross section of the manifold. The rest of the KL computation is equivalent to the process described in Section 4. Here we chose this evaluation method over calculation of marginal statistics (as in Fig. 1G and Fig. S3) because some models produced samples far away from the ground truth manifold. Models with stronger L1 regularization generally showed lower performance across most layer depths (Fig. S2). Skip connections have a more nuanced effect. In deeper networks, they result in large performance gains when compared to models without skip connections. However, in shallower networks, skip connections result in comparable or sometimes worse performance than their counterparts with no skip connections.

Why is this the case? Skip connections such as those used by deep networks allow for better performance by preventing vanishing/exploding gradients; by giving the network another way to route information, they induce a smoother loss landscape that is easier to traverse during training (Li et al., 2018). Even though our skip connections have learnable parameters, information can still be routed along different pathways, allowing for more direct influence over particular subunits. This effect is likely more pronounced in our tree architecture than in models with fully connected layers since there is greater segregation of information at each layer. The relative loss of performance in shallower networks suggests skip connections may induce destructive interference that cannot be remedied if there are not enough downstream layers for further processing. Indeed, we see that when the model is L1 regularized and has skip connections, optimal performance occurs when the architecture is neither too shallow nor too deep, but rather possesses some intermediate number of

layers. This may have implications on the type of dendritic architectures we expect to see in sensory cortices.

## B.2 TESTING NEURAL SAMPLING QUALITY BY MARGINAL STATISTICS

The data manifold on which the model in Fig. 1A was trained has three embedding dimensions but is embedded in a 10-dimensional ambient space (using the terminology in Jazayeri & Ostojic (2021)). We ensured the data have non-zero values in every dimension by rotating the three dimensional manifold by $\pi/4$ around each pair of axes.

The three dimensional swiss-roll manifold is uniformly distributed along the linear dimension (i.e. the dimension orthogonal to the swiss-roll cross-section). We evaluated the marginal statistics of the model output along this dimension by comparing the cumulative distribution function (cdf) of this model output against a reference uniform cdf.

To recover the model's marginal density along the linear dimension, we first projected the network output onto the three basis vectors of the embedding space, which we calculated by transforming unit vectors by the same rotation operators as the data manifold. We then projected the model output along the third basis vector and computed the cdf along this dimension.

The empirical cdf is close to the reference cdf, indicating that the data are likely distributed according to the uniform distribution. We can quantify this using the Kolmogorov-Smirnov goodness-of-fit test, which is a nonparametric measure of the similarity between a continuous empirical sample distribution and a specified reference distribution. We applied the KS test to both the empirical distribution along the nonlinear dimension in Fig. 1G and along the linear dimension in Fig. S3. For a sample size of 1000 each, the p-value along the nonlinear dimension is $0.145$ while the p-value along the linear dimension is $0.125$, making them statistically indistinguishable from the uniform distribution.

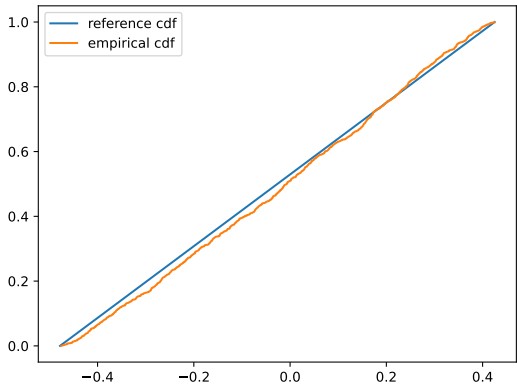

Figure S3: Marginal statistics along the dimension orthogonal to the swiss roll cross-section.

## B.3 POSTERIOR AUTOCORRELATION

We numerically evaluated posterior samples generated by the neural circuit. We see in Fig. S4 that the samples generated from neural sampling result in autocorrelation functions that rapidly decrease and remains steadily at zero. This indicates that the posterior samples, like the prior samples in Fig. 1G, are largely independent over time despite the approximations. This remains true even when the neural sampling dynamics are influenced by contextual and sensory evidence.

## B.4 STOCHASTIC NEURAL NETWORK

To assess whether the nonlinear operations of the neural sampling process could be instantiated by a population of neurons rather than in the dendritic arbor of pyramidal neurons, we trained a Stochastic Neural Network (SNN) model, a fully connected feedforward neural network comprising neurons with linear weights, ReLU nonlinearity and somatic stochasticity. As in our dendritic model, the

**A**

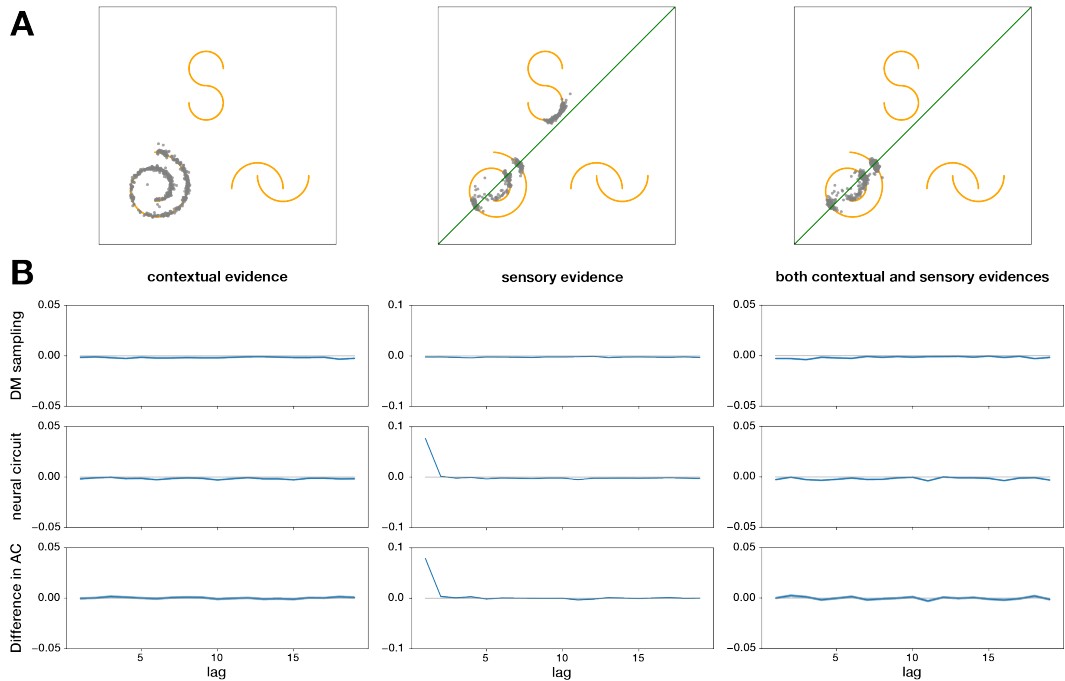

Figure S4: Autocorrelation comparison of the posterior samples. **(A)** Posterior samples generated from the two dimensional trimodal dataset as a result of observing contextual evidence (left), sensory evidence (middle), and both forms of contextual and sensory evidence (right). The context cue for the context-conditioned posterior is the discrete context random variable $c_2$ indicating the swiss roll manifold. The sensory likelihood, as before, is a two dimensional linear constraint with a Gaussian-distributed uncertainty around it. **(B)** The autocorrelation functions of samples generated by the DM reverse process (top row) and neural sampling (middle row) for the different posterior distributions. The difference between the two autocorrelation functions is shown in the bottom row.

somatic stochasticity in each neuron is additive Gaussian, with the degree of stochasticity modulated by the phase of the global oscillation. The operation performed by the neurons in each hidden layer can be expressed as

$$h_l = \text{ReLU}(W_l(h_{l-1}, t)) + (1 - \beta_t)\epsilon_t, \tag{7}$$

where $h_l$ is the output of hidden layer $l$ and $W_l$ is the weight matrix at layer $l$. $\epsilon_t$ is a sample of a Gaussian with identity covariance.

Our instantiation of this model architecture has three hidden layers, each with a width of 32 neurons. The degree of noise as given by the oscillator $\beta_t$ is concatenated with the output of the previous hidden layer. We trained the model via backpropagation using the Adam optimizer with a learning rate of 1e-4 over 5000 epochs. We show the distribution of samples generated at intermediate points of the ascending phase below, in Fig. S5.

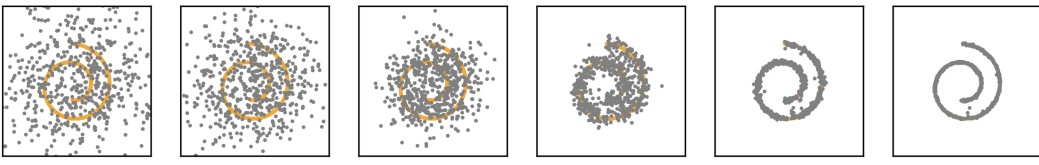

Figure S5: DM reverse process using the SNN architecture. The samples generated by the SNN result in samples that lie on the two dimensional swiss roll manifold.

### B.5 More biologically realistic oscillations

In the brain, oscillations fluctuate in both amplitude and frequency and do not have the perfect sinusoid structure assumed by the model. To test the effects of such fluctuations on the circuit's

ability to sample from priors, we introduced variability in the amplitudes and/or period of the $\beta_t$ oscillation. The original oscillation cycled between 0 and 1 with a period of 200 timesteps (Fig. S6A). In these variations, the amplitude and period of the oscillation were defined as random variables. The amplitude was drawn from a truncated normal distribution $A \sim \mathcal{N}(A_{\text{original}}, \sigma_A), A > 0$, i.i.d for each cycle (Fig. S6B), where samples were obtained by rejection sampling. The period of each cycle was similarly drawn from another truncated normal distribution $T \sim \mathcal{N}(T_{\text{original}}, \sigma_T), T > 0$ (Fig. S6C). Finally, we also considered the combined effect of simultaneous fluctuations in both amplitude and frequency (Fig. S6D). Using the denoising-trained 2D swiss-roll model, we used the induced $\beta_t$ values for circuit sampling. We assessed the effect of the generated samples by comparing the empirical distribution given by the resulting samples against the ground truth distribution, via the KL divergence. We find that the quality of the samples is largely unaffected by the deviations from the perfect sinusoid (Fig. S6E). The KL divergence remains largely flat for a range of noise levels. This suggests that the sampling procedure can be reasonably robust for naturalistic oscillatory signals.

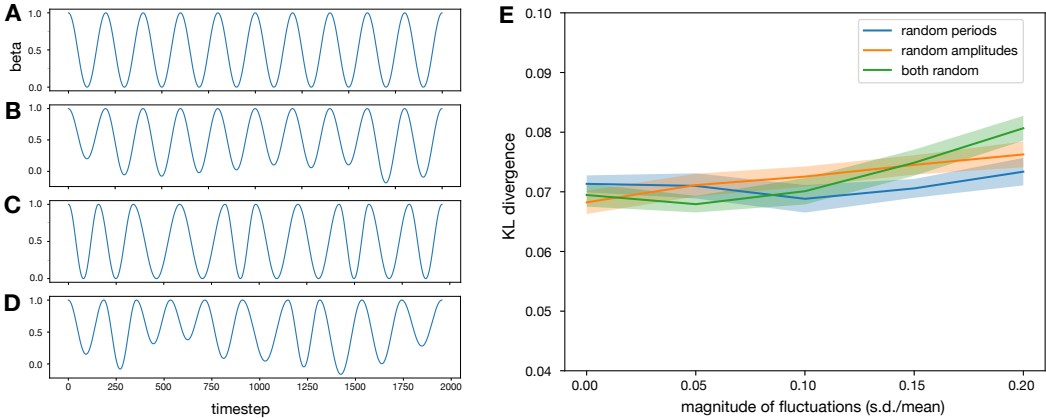

Figure S6: **(A)** Reference $\beta_t$ fluctuations for the simple model. **(B)** Example draw for a more naturalistic oscillations with fluctuations in amplitude ($\sigma_A = 0.15$). **(C)** Example draw for a more naturalistic oscillations with fluctuations in period ($\sigma_T = 30$). **(D)** Example draw for a more naturalistic oscillations with fluctuations in amplitude ($\sigma_A = 0.15$ and $\sigma_T = 30$). **(E)** Deviations of empirical samples from ground truth distribution for varying degrees of oscillation fluctuations. A standard deviation corresponds to the deterministic setting. KL estimates use 5000 samples per run, with shading showing SEM estimated over 20 runs.

## B.6 MNIST DATASET

To assess whether our model scales to higher dimensional priors, we trained our circuit model on the MNIST dataset, which contains 28 by 28 pixel images of handwritten digits. We were able to train several dendritic architectures to successfully generate digits from the MNIST dataset (Fig.S7, top two rows). They were trained by the same denoising objective, using the Adam optimizer with a learning rate of 4e-3 for 5000 epochs. The best performing architectures were two and three layer architectures with branch factors $[12, 12]$ and $[5, 5, 5]$ respectively.

We sampled a class-conditioned posterior by combining the model with a separate classifier trained on MNIST digits (Fig.S7, top two rows). We used approximate posterior dynamics described in Section 3 and described in Supplementary section A.1. Taken together, these MNIST results suggest that the biologically-motivated deviations from traditional diffusion models can scale to higher dimensional distributions.

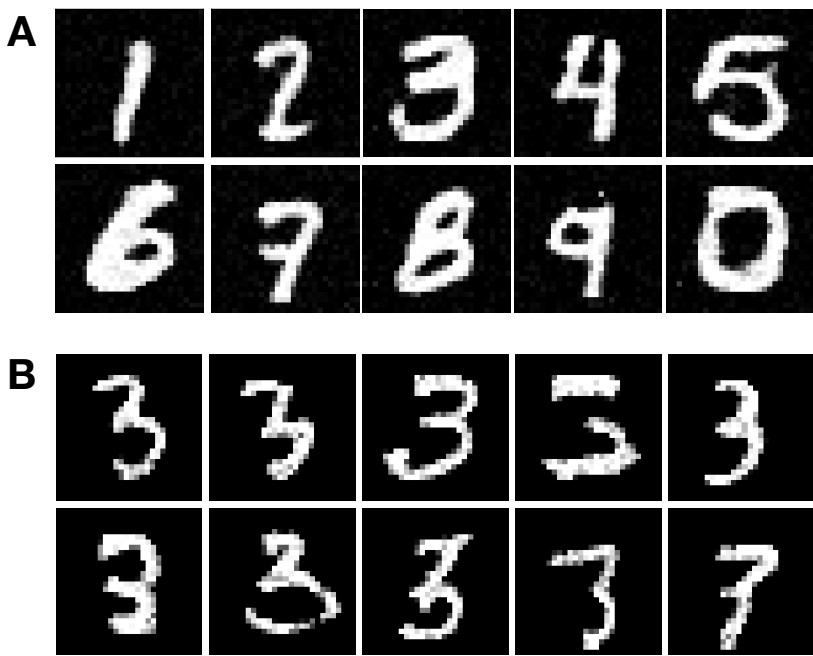

Figure S7: Results for an an MNIST trained neural circuit. (**A**) Example prior samples, one per class. (**B**) Example samples from the posterior conditioned on class '3'.

