# OpenReview forum: "Complex priors and flexible inference in recurrent circuits with dendritic nonlinearities"
_ICLR.cc/2024/Conference — ICLR 2024 spotlight_

### Official Review · Reviewer_A1Uf · 2023-10-28

**Soundness:** 3 good
**Presentation:** 3 good
**Contribution:** 2 fair
**Rating:** 6
**Confidence:** 2

**Summary:**

The paper introduces a novel recurrent circuit model based on diffusion models that incorporates several biologically plausible properties. This method implicitly encodes priors over latent variables and can combine this information with other sources, such as sensory input, to encode task-related posteriors. The approach is mapped to a recurrent network with multi-compartment neuron models, and its effectiveness is demonstrated through experiments on toy datasets.

**Strengths:**

* The paper successfully maps diffusion model-inspired dynamics to a recurrent circuit with multi-compartment neuron models,

* The modifications to diffusion model dynamics to align with biologically plausible properties are commendable, especially as they do not harm the network quality,

* The provided code is clear and enhances the paper's reproducibility.

**Weaknesses:**

I think the paper has several weaknesses. Please see the following list and the questions sections.

* Regarding the following sentence in page 5: "First, we tested ... autocorrelation ... remains steadily around zero proving that ... samples are essentially independent." I might agree that the samples are independent. However, isn't it mathematically misleading to use the word "prove" based on correlation?

* Regarding the sentence in page 5: "Overall, these results indicate that the constraints imposed by biology may have a minimal effect on the quality ... compared to DMs". Isn't it too early to state such conclusion. The experiment is on a quite toy-dataset.

* The experimental details for the stochastic neural network in Appendix B.2 are lacking, and providing more comprehensive information in this section would strengthen the paper.

* Overall, the experiments are on quite toy datasets.


**Minor Comments**

*  In the second line of Section 2, "nosy" should be corrected to "noisy."

* Figure 1 lacks a caption for (F).

*  In Figure 4 caption (for (B)), it is written "(left) ... (right)" to point out to the corresponding plots, but I guess the caption should use "(top)" and "(bottom)" to indicate the corresponding plots.

* The first line of Section B.3: "Fig. ??" which figure is that?

**Questions:**

* In Figure 1D, the caption mentions "multi-compartment neuron." Shouldn't it be "multi-compartment neurons"? I thought each gray triangle is a multi-compartment neuron.

* The paper repeatedly mentions "optimized ReLU nonlinearities." I did not really understand the meaning of "optimized" in them?

* Is $\sigma_\lambda$ in Equation 1 the same with $\beta_\lambda$ in Figure 1?

* Regarding the following sentence in page 6: " In principle, this might lead to catastrophic accumulation of errors and large sampling biases; however, the attractor dynamics prevent this from happening." Can authors elaborate on this a bit more? It is not clear to me why?

* In page 6, it is written "$\mathbf{g}\_t (\mathbf{x}\_{t-1}, s) = \frac{1}{\sigma_s^2} \mathbf{M}_s \mathbf{M}\_s^T (\mathbf{x} - \mathbf{x}\_c)$". Is it $\mathbf{g}\_t (\mathbf{x}\_{t-1}, s) = \frac{1}{\sigma\_s^2} \mathbf{M}\_s \mathbf{M}\_s^T (\mathbf{x}\_{t-1} - \mathbf{x}\_c)$?

* Does the last sentence of Figure 3 caption correspond to Figure 3D? There is no caption for (D).

---

> ### Author Response · Authors · 2023-11-23
> **Reply reviewer A1Uf**
>
> Thank you for the positive overall evaluation of the work.
> We agree that numerically validating that the autocorrelation of the RNN-generated samples decaying to zero at small lags does not mathematically ‘prove’ that this will be the case in all conditions; the same goes when demonstrating that the additional biological constraints do not harm sampling performance. We can still say that –empirically–  it is the case across the distributions that we have tried (including new results on MNIST), both for the prior and the posterior samples.
>
>
> Regarding the role of attractor dynamics in correcting for the approximations, one way to think about the approximations introduced in the service of biological plausibility is as inducing small noisy deviations from what would have been the ideal trajectory, defined by the solution without the approximations. Since in our model the representation of distributions in the embedding manifolds is driven by the attractor flow fields, adding moment by moment stochasticity on top of such dynamics can be partially corrected for by the push towards the manifold. The dynamics will not end up exactly at the same location on the manifold compared to the exact version, but they will still sample from the right distribution.
> About the use of toy datasets: see common reply. Briefly, we now numerically demonstrate the scalability of our approach using MNIST.
>
>
> We have corrected the minor typos and so on (see updated pdf).
> Questions:
> - Multi-compartment neuron should be plural (typo).
> - We used the phrase “optimized ReLU nonlinearities” to mean that the dendritic nonlinearities have been optimized on the denoising objective defined by Eq. (4). These nonlinearities are functionally instantiated by linear weight matrices and ReLU activations. We have modified the main text for clarity.
> - It is indeed the case that $\sigma_\lambda$ in Eq 1 is the same as $\beta_\lambda$ in Fig. 1. To keep our notation consistent, we changed Fig 1 to use $\sigma_\lambda$ instead.
> - This sentence relates to our approximation of the likelihood, which consists of evaluating $g_t$ at $x_{t-1}$ rather than at $\mu_\theta(x_{t-1})$. There is a substantial penalty associated with this approximation, since $x_{t-1}$ is the noisy sample at time $t-1$, while $\mu_\theta(x_{t-1})$ is the network’s (denoised) estimate of the clean sample at time $t$, which lies closer to the manifold of clean images. Since this approximation error is incurred at every timestep, it is possible for these errors to accumulate over time and lead to large sampling biases in the resulting posterior distribution. Instead, we find in our numerical simulations that this is not the case; rather, the discrepancy (measured using the KL divergence) tends to decrease over the course of the inference process.
> - Yes, this is correct. We have fixed the typo to include the time step in $\mathbf{g}$.
> - The last sentence of the caption of Figure 3 does indeed correspond to Figure 3D. We have fixed the caption.

---

### Official Review · Reviewer_7hMe · 2023-10-30

**Soundness:** 2 fair
**Presentation:** 2 fair
**Contribution:** 2 fair
**Rating:** 5
**Confidence:** 2

**Summary:**

This paper proposes the state-of-the-art diffusive model developed in deep learning could be a running algorithm in recurrent neural circuit. Specifically, it claims the nonlinear dendrites of neurons with globally controlled dendritic noises can be used to implement the reverse phase of diffusive models.

**Strengths:**

The concept of the present study is original and can significantly advance our understanding of the stochastic recurrent neural circuit once some of my major concerns are solved (see weaknesses). The structure of the manuscript is organized well, and the introduction and discussion are clearly written with a thorough review of the research history as well as possible experimental verification of the theory.

**Weaknesses:**

I have major concerns about the justification and derivation of two central claims (dendritic nonlinear and global oscillating signal) about neural circuit implementation of diffusive models. A possibility is that the text doesn't explain the key math steps sufficiently well. I look forward to seeing some justification in the rebuttal.

### Dendritic nonlinearity
The paper directly states the nonlinear dendritic operation $f(x_t, \beta_t)$ after Eq. 5 without explaining how it is derived. I am confused about how this nonlinearity comes out. Is it directly from $\mu_\theta (x_{\lambda + t}, \lambda)$ in Eq. 2? In this case, does it imply the dendritic nonlinearity needs to be readjusted if the transition operator in the reverse process is changing? Or it is just used as a way to capture the nonlinearity in biological neurons?

### Global oscillating signal
I don't understand how the global oscillating signal is derived. Although the author explained the $\beta_t$ is analogous to the sequence of noise variance in the diffusive model, it seems that the diffusive model doesn't have such an oscillating signal if I understood correctly. I have no idea how Eq. 5 was derived, what assumptions it relies on, and why the $\beta_t$ becomes a sinusoid function there.

### Training an ANN-based model for dendrites
The author says the dendrite is modeled as an ANN whose parameters were trained via gradient descent. Does this imply some mechanism to adjust the dendritic parameters in real neurons? If so, what are the possible biological mechanisms? I don't see related discussions in the paper.

Overall, I suggest the author explain the key motivations, and assumptions in deriving the nonlinear dendrite, global oscillating signals in modulating dendritic noises around Eq. 5. They seem quite critical in this work. In addition, it is better to write a concrete, commonly used recurrent neural dynamics in a single line, and compare it directly with the reverse process in diffusive models. If my major concerns are solved, I'd like to increase my rating.

**Questions:**

- Fig. 1E caption: why does neural dynamics push off the network off manifold? Does the author mean the neural dynamics of sensory transmission correspond to injecting noise in a similar fashion with the forward process in a diffusive model? At least this neural dynamics is not the same recurrent neural dynamics which the author claims to implement a reverse process to sample from posteriors.

- Fig. 4B caption: it should be (top) and (bottom) because no (left) and (right) here.

- It seems that the iterative steps in diffusive model are indexed by $\lambda$ while that in the neural dynamics was indexed by time $t$. What's the relation between $\lambda$ and $t$? My understanding is that $\lambda$ is a non-negative number while $t$ can go to infinity. Does it imply the equilibrium neural dynamics repeatedly sample distribution of $x_0$ over time?

---

> ### Author Response · Authors · 2023-11-23
> **Reply reviewer 7hMe**
>
> We thank the reviewer for recognizing the originality of our work and we apologize for the lack of clarity in the description of the model.
>
> Dendritic nonlinearities: The collection of single neuron nonlinearities (which have a shared parametric form, but with neuron-specific parameters), are the direct result of the denoising optimization procedure, and their operations map one to one onto  the reverse transition operator $\mu_\theta(x_{\lambda+1}, \lambda)$ in  Eq. 2.  This indeed implies that the dendritic nonlinearity parameters would change when modeling a different prior distribution (biologically achievable through a combination of synaptic and branch plasticity, although we do not model the mechanics of learning here).
>
> Global oscillations: there is a one-to-one map between the oscillation phase in our model  and the noise schedule in the traditional diffusion models at sampling time. In more detail: in diffusion models, the degree of noise at step $\lambda$ is given by a noise schedule $\sigma_\lambda$ that decreases monotonically over the course of inference as a noisy image $x_\lambda$ is iteratively denoised. In other words, images with the largest degree of noise are found at $\lambda=\Lambda$ and clean images are produced at $\lambda=0$. While the exact form of the schedule can matter in the details, there is a range of possible choices available. In the interest of biology we used a sinusoidal interpolation between  $\lambda=\Lambda$  and  $\lambda=0$. This allows us to establish a direct map between  $\Lambda$  and the ascending phase of a circuit oscillation (see Eq.5), such that images with the largest and smallest degrees of noise correspond to the trough and peak of the oscillation, respectively (this is a somewhat arbitrary choice, motivated by some of the past literature). There is no direct correspondence to the descending phase of the $\beta_t$ oscillation in diffusion models, which is a unique element of our solution. The traditional forward process would provably achieve the correct outcome computationally (since its entire purpose is to map the actual input distribution into a gaussian), something which is not formally guaranteed for the oscillation. However, that solution would not match the biological constraint that the neural dynamics driving the evolution of network activity cannot dramatically change every few ms or tens of ms, whereas we can use the same RNN dynamics with increasing levels of noise and keep things consistent.
>
> Training an ANN-based model for dendrites: the biological implementation of this process is beyond the scope of this paper, but there are some technical results that suggest that one can define a local denoising objective that does not require knowledge of the clean image, but only of the the general noise statistics used to corrupt it (Raphan and Simoncelli, 2014), so it is possible in principle to construct a denoising optimization procedure that involves relatively local computations. To which extent those operations map to experimental observations on synaptic and branch plasticity remains to be determined.
>
> Question answers:
> -in Fig.1E The only difference between the ascending and descending phases is whether the degree of noise that is present in the neural transition operator is decreasing or increasing. The increasing level of noise in the dynamics effectively drives the activity pushing it increasingly off manifold. The statistics of exactly where one ends up are not provably identical to those imposed by the traditional feedforward process, but when interacting with the attractor dynamics in the ascending oscillation phase it seems to be close enough to effectively decorrelate samples across cycles.
> - sorry for the misdirection, we have now corrected the caption to top/bottom
> - we made the choice of indexing the network by time but the nature of the operations really depends by the phase of the oscillation, which maps one to one to the time step within a cycle in regular diffusion. We were hoping that it makes the correspondence between what happens algorithmically and what might happen in the brain clearer, but it does add extra notation, perhaps unnecessarily.

---

### Official Review · Reviewer_XKTM · 2023-10-31

**Soundness:** 3 good
**Presentation:** 3 good
**Contribution:** 3 good
**Rating:** 6
**Confidence:** 3

**Summary:**

In this paper, the authors proposed a diffusion-based recurrent circuit for sampling-based probabilistic computation. They used recurrent circuits to represent complex priors implicitly, and the sampling-based inference was accomplished using noise modulated by an oscillatory global signal. The recurrent circuits implement the diffusion models with dendritic nonlinearities and stochastic somatic integration. They showed that the dynamics can be gated by bottom-up or top-down signals to generate samples from the corresponding posteriors in low-dimensional nonlinear manifolds and multimodal posteriors to achieve flexible inference.

**Strengths:**

There have been earlier works exploring ideas of recurrent connections to encode priors and the neural dynamics to perform Langevin sampling. The proposed plausible neural circuit implementation of across-task inference in which a common prior is encoded in the recurrent connections with dendritic nonlinearities optimized for denoising is novel.  The connection to diffusion models,  the use of global oscillatory signals as sampling control, and the use of bottom-up and top-down signals for gating the samplings across multiple tasks are also new and interesting and represent a conceptual advance. It does provide a new plausible framework to allow flexible sampling of complex distributions.

**Weaknesses:**

While the connection to DM is inspiring, there is no direct evidence to support the key assumptions and innovation of the model -- the dendritic nonlinearities and DM-inspired oscillatory sampling schedule. They remain a fragment of imagination.  As it is mostly a theoretical neuroscience model that works only on a toy example for demonstration, it would be worthwhile to articulate the predictions and the assumptions of the model that can be tested and evaluated by neurophysiological experiments.

**Questions:**

How can the models be tested? What evidence would prove they are correct or falsify them?

---

> ### Author Response · Authors · 2023-11-23
> **Individual reply Reviewer XKTM**
>
> We thank the reviewer for the positive comments.
>
> It may be worth noting that our sampling dynamics are qualitatively quite different from Langevin, even though it is indeed the case that our work builds upon a body of work on MCMC approaches for neural sampling.
>
> Providing direct evidence for individual elements of our solution is indeed difficult, but not outside the scope of future experiments. There is ample indirect evidence that similar kinds of computations are possible in the brain: certain interneurons types have both oscillatory responses and target principal cells’ dendritic arbors, which means that they can modulate the effective neuron nonlinearities in a cyclic fashion. Similarly, the phase of the local oscillation can affect somatic subthreshold variability and change the noisiness of the generated neuron outputs. Finally, there is ample evidence (at least from within the hippocampus and for hippocampal-cortical interactions) that across area communication happens preferentially at certain phases of a local oscillation, and Communication through coherence theories have made similar arguments for the cortex. So, we would argue that all of the individual elements of the model have at least loose experimental support (as in something of that flavor is known to happen, even if it cannot be matched in the details).  That said, we do identify one very concrete analysis that would prove or disprove the model in the paper, in the form of activity pattern similarities across different phases of an oscillatory cycle. There is experimental proof of feasibility of such an analysis (used for a slightly different purpose) in the work of Berkes et al.
>
> About the use of toy example models: see common reply. Briefly, the low dimensionality of the distributions considered is a convenient choice rather than a limitation of the model; we now include a MNIST experiment representing actual images to illustrate that point.

---

### Official Review · Reviewer_8Gad · 2023-11-01

**Soundness:** 3 good
**Presentation:** 3 good
**Contribution:** 2 fair
**Rating:** 6
**Confidence:** 3

**Summary:**

This paper presents a neural circuit model for Bayesian inference, where the prior is proposed to be encoded via a DDPM-like mechanism in the dendrites of neurons. Some additional modifications were made to be more biorealistic, such as the dendrites instantiated as tree-structured MLPs, and iterative prior sampling driven by a global oscillatory noise schedule (as opposed to the standard, “discontinuous” reverse diffusion sampling). The authors show demonstrate that, on a swiss-roll toy task, the neural sampler performs similarly to the standard DDPM. Furthermore, the authors demonstrate how the prior can be flexibly combined with multiple additional sources of information (likelihoods), such as sensory observation and contextual signals, for posterior sampling. Finally, the authors make experimental predictions on how different phases of the global oscillation would be informative if such a representation is implemented in real neural circuits.

**Strengths:**

I found the paper to be clearly and concisely written, with very informative figures to illustrate the main idea and key results. The proposed circuit implementation of a diffusion model in the dendrites is interesting and novel, while the oscillatory noise schedule is an elegant device, with an intuitive (though potentially misleading) mapping to neural oscillations. As demonstrated, the proposed model can incorporate multiple likelihood sources separately, and at least for the toy tasks, achieve good performance. I did not examine the math thoroughly, but overall I believe the paper is of high technical quality, and presents an interesting hypothesis (and testable model) for how neural circuits may encode priors.

**Weaknesses:**

For me, the paper suffers from two major weaknesses:

First, while the model is elegant and combines a SOTA class of generative model in ML (DDPMs) with neural circuitry, the end product feels too artificial in construction and biologically implausible, with its many strong restrictions / assumptions. For one, the diffusion model in this implementation can be trained as per usual, but how would real dendrites in a neuron go through this learning? The authors state the the learning aspect is for future work, but that’s a huge “if”—is it really realistic to suppose that such a complex mechanism can emerge in vivo, without any hint for the readers of how it could? Additionally, sampling critically depends on a stationary and permanent oscillation, which is rarely found in the brain. Even hippocampal theta, which the authors draw inspiration from, are irregular in time and frequency, nevermind oscillations in other cortical areas.

Second, the performance is only demonstrated on a relatively simple task of sampling from a very smooth and low-dimensional manifold. I understand that the quantitative results are mostly a demonstration of proof of principle. However, there’s no indication that this would work with mildly more complex high-d distributions, as the authors had originally motivated (and pointed out as a weakness in previous literatures, i.e., mostly Gaussian prior/posteriors). It would be nice to have some indication of how this can be scaled to perform a mildly more complicated task, like conditionally sampling MNIST, or would it be completely infeasible given the architectural constraints?

These are the two major categories of concern, and I have a number of other concrete issues in the limitation / questions sections below. Taken together, I am skeptical of how much of the claims regarding “neural circuit implementation” is substantiated. And if not, how impactful would the contribution be, which is essentially connectivity-constrained DDPM with an oscillatory noise schedule. Therefore, I recommend borderline reject, noting that it is a well written paper (with some technically dense sections) and solid work but perhaps for a more niche readership, and that I am open to be convinced of its potential biological relevance.

**Questions:**

- is there recurrent interaction between neurons? It’s also a bit unclear how many neurons there are, or are all the results from a single dendritic tree? Also, what exactly are the inputs/outputs of the dendritic networks, and what exactly is the somatic “compartment” doing, or is that functionally just the last layer of the dendrites? Apologies if I had missed this obvious info.
- In the case of posterior inference (Figure 2/3), do the samples still show zero autocorrelation?
- I may have fundamentally misunderstood something, but the authors motivate their proposed architecture as flexible, since it can be reused for various inference scenarios, just swapping out or combining likelihoods. This is demonstrated well for the current model, but wouldn’t this be true for a model where a different population encodes the prior as well? Why is it necessary that it’s in the dendrites? Or is that just a “semantic” difference, since the branching networks can equally be implemented as different neurons?
- Isn’t it typically the case that bottom-up sensory info and contextual info are thought to be likelihood and prior, respectively? Whereas here, they are represented as two different steams of likelihoods. Can the authors comment on this discrepancy?
- there is a recent body of experimental evidence implicating oscillations of different frequencies coordinating bottom up (gamma, ~40Hz) vs. top down (beta, ~15Hz) signaling (see A. Bastos, EK Miller, etc.), while here it’s crucial that there is a global oscillation of a single frequency, otherwise the prior and likelihood sampling are temporally misaligned.
- The proposed implementation draws one prior / posterior sample per oscillation cycle, which, given the fastest cortical rhythm (40Hz gamma), results in 40 samples per second, and more likely to be less, e.g., 8Hz theta in the hippocampus. Is this sampling speed too slow? Does it match behavioral data of evidence accumulation?
- As I mentioned above, most of the time, most areas of the cortex are not experiencing oscillations. Furthermore, oscillations tend to disappear during task engagement (such as 10Hz alpha in visual areas and 20Hz beta in motor areas), which would be a time that is critical for sampling. How reliable would the proposed implementation be in such scenarios?
- caption for Figure 1F is missing

---

> ### Author Response · Authors · 2023-11-23
> **Individual reply Reviewer 8GAd**
>
> Thank you for the feedback.
> About the general biological plausibility of our scheme (see also reply to reviewer XKTM):
>
> Oscillation: It is fundamentally in the nature of modeling to abstract away some of the inessential details. Our oscillatory signal is such an abstraction. In new results we now show numerically that the core mechanism remains largely unaffected when using an imperfect oscillation, which varies in amplitude and period from cycle to cycle by a significant margin (Suppl. Section B5).
>
> Learning: Learning probabilistic representations is a huge open question in the field. Virtually none of the published circuit level models provide a learning procedure that can construct them. This is true about models of priors (Ganguli and Simoncelli, 2014, although something similar can be learned with a REINFORCE type of rule, Bredenberg et al, 2020) and -to our knowledge- all neural sampling literature. That said, denoising as a learning objective is more local and in some ways mathematically simpler, which makes us hopeful about our ability to map it into plasticity-based biological learning. Indeed, there are some technical results that suggest that one can learn a Bayesian estimator without requiring knowledge of the distribution of clean data, but only of the the general noise statistics used to corrupt it (Raphan and Simoncelli, 2007), which suggests a mechanism by which biological systems may learn to denoise without having ever seen clean data.
>
>
> About task complexity: see common reply. Briefly, the distributions we consider are actually quite complex in light of the previous neural sampling literature (complexity here is defined as nonlinear low-d manifolds embedded in high-d ambient spaces and strong multimodality). None of the previously published neural sampling models would be able to represent or sample from our ‘simple’ toy examples, so this is already a qualitative improvement over the comp. neuro. SOTA. Further, we now include new experiments with MNIST to show that the model can be scaled up without problems (Suppl.Section B6).
>
> A more detailed note on the limits of distributional complexity that can be in principle represented: there is a natural question about “how complex can it possibly get?” Based on the machine learning literature on diffusion models the answer is at least pixel level natural images, although the number of parameters and complexity of dendritic nonlinearities involved in a naive implementation of that is unrealistically high. In subsequent work we are exploring how hierarchical versions of the same idea would scale up prior complexity while keeping the neural resources required for its representation relatively small.
>
> Detailed question answers:
>
> Recurrence: indeed this is a network of all-to-all connected units, each of which involves a neuron specific nonlinearity, with parameters trained by denoising.  The number of neurons is the same as the ambient space (10). The somatic compartment has a somewhat privileged role in that it introduces stochasticity in the neuron responses (although we have tried a version where noise was injected along the dendritic tree, with no particular ill effect). These details are now clearly explained in the updated manuscript.
>
> Autocorrelation function of posterior: Yes, the samples still show zero autocorrelation for posterior inference. The results are shown in supplement, in the section titled B.3 Posterior Autocorrelation.
>
> Flexible inference in other schemes: In principle, a linear PPC circuit that separately encodes the log prior can combine its representation additively with incoming sensory/contextual log likelihoods to perform inference; flexible inference would require a different kind of gating between subpopulations, but it can in principle be done, although to our knowledge it hasn’t been demonstrated explicitly. Nonetheless, this only works for one-dimensional or fully factorized joint distributions. The type of priors the brain needs to represent are arguably more complex which has motivated probabilistic circuits based on sampling. We now provide a way to achieve flexible inference for such complex distributions.
>
> Should the top-down contribution be called ‘prior’ or ‘likelihood’: We agree that our choice of wording is (although formally correct) a little inconsistent with some of the neural literature. We are thinking about these qualifiers from the perspective of an intermediate stage of a hierarchical inference process, and so the definition of the role of top-down information is with regards to the locally represented latents.

---

> > ### Author Response · Authors · 2023-11-23
> > **Continued...**
> >
> > Multiple interacting oscillatory bands: thank you for the references. At this stage we don’t have a computational interpretation for the role of multiple oscillatory bands in interarea information exchange. Worth thinking about.
> >
> > Theta vs. gamma: the model does not really constrain the band. We motivate some model choices based on hippocampal results (theta frequency) but in the cortex (lower) gamma is a more likely candidate. I would argue that it is generally unclear what is the rate at which sampling would need to happen to account for perception (although I know that at some point there were experiments in the Legyel/Fiser groups trying to estimate that).
> >
> > Modulation of slow oscillations by behavioral context: it is not clear how those empirical observations relate to what we are talking about here, but it is likely a separate process. More generally, the global coordination of information across the full visual hierarchy is very interesting, and something we are actively working on right now but it goes beyond the scope of this particular paper.
> >
> > Thanks for pointing out the missing caption. Now fixed.

---

> > > ### Comment · Reviewer_8Gad · 2023-12-05
> > >
> > > Thanks to the authors for their detailed response. While it is true that DDPM is a very flexible class of model that, in theory, would not lose expressiveness when embedded in such a bio-realistic set up, it is still good to see that it works on MNIST. I still have concerns regarding the realism of the oscillatory noise schedule and how it limits sampling speed, but I appreciate the additional experiments on robustness to non-stationarity. Overall, I think it's a nice piece of work that is of high enough quality to be presented, and therefore updated my score from 5 to 6.

---

### Author Response · Authors · 2023-11-23
**Common reply on the complexity of the modeled distributions**

We thank all reviewers for their comments.

There was a general concern expressed across reviewers that the examples included were too “toy” level, and that it was unclear how the model would generalize to higher dimensional natural scenarios.

First, we do know from the machine learning literature that diffusion models have the capacity to represent very complex distributions.
Second, the toy examples chosen allow for not only a simple visualization of the produced samples but also precise quantification of sampling quality with respect to the ground truth distribution, something which would be difficult for general tasks.
Third, we now include new numerical experiments with pixel-level prior representations of MNIST images, as evidence that scaling the idea up to higher dimensional problems is possible and that the biological constraints do not incur substantial costs in terms of the quality of the generated samples even in that regime.
Finally, one should also note that even the toy examples presented here would pose significant challenges if one were to try to represent them with any of the previously published probabilistic coding schemes; in those terms, our approach is a qualitative leap over previous neural sampling models in the complexity of representable distributions (past work is almost exclusively restricted to multivariate gaussians).

---

### Meta-Review · Area_Chair_f9Kg · 2023-12-08

**Metareview:**

The paper describes a biophysically plausible (claimed) learning procedure for implicit representations of priors in a recurrent neural network based on a de-noising objective as used e.g. in diffusion models (DMs).

The reviewers generally all appreciated the innovation, quality, and presentation of the proposed model and its validation. Likewise, their criticism was aligned and focussed on the claimed neuro-realism for probabilistic computations in the brain. In particular two aspects were repeatedly argued in that regard: how non-linear dentritic integration optimized for de-noising can be learned, and the need of a global 'clock' to modulate the signal noise. The authors did a good job in discussing those points with the reviewer, swaying some of the concerns but without being able to fully eradicate them.

The ACs impression is that this is an interesting paper describing an interesting network and learning architecture for performing probabilistic inference of tasks that rely on naturalistic and complex structured priors. The quality of the work and the clarity of the explanations is exemplary. However, the authors did somewhat shoot themselves in the foot with making the claims about the framework being a realistic proposal for how biological neural networks learn to and perform inference. None of the assumptions seem implausible but  claims need to be defendable and that is not (yet) the case for the proposed model. Thus the AC shares the sentiment and skepticism of the reviewers in this regard. However, overall a high quality and interesting paper.

**Justification For Why Not Higher Score:**

The concerns about the claimed biophysical realism is the main reason for not scoring it higher.

**Justification For Why Not Lower Score:**

It is a quality paper proposing an interesting solution to an important problem: how neural circuits can implicitly learn complex structured priors from natural input signals, and can perform probabilistic inference across different tasks.

---

### Decision · Program_Chairs · 2024-01-16

Accept (spotlight)